# Transcriptome Analysis Provides Insights into the Mechanism of Astaxanthin Enrichment in a Mutant of the Ridgetail White Prawn *Exopalaemon carinicauda*

**DOI:** 10.3390/genes12050618

**Published:** 2021-04-21

**Authors:** Yue Jin, Shihao Li, Yang Yu, Chengsong Zhang, Xiaojun Zhang, Fuhua Li

**Affiliations:** 1Key Laboratory of Experimental Marine Biology, Institute of Oceanology, Chinese Academy of Sciences, Qingdao 266071, China; jinyue@qdio.ac.cn (Y.J.); yuyang@qdio.ac.cn (Y.Y.); chs-zhang@163.com (C.Z.); xjzhang@qdio.ac.cn (X.Z.); 2Laboratory for Marine Biology and Biotechnology, Qingdao National Laboratory for Marine Science and Technology, Qingdao 266000, China; 3University of Chinese Academy of Sciences, Beijing 100049, China; 4Center for Ocean Mega-Science, Chinese Academy of Sciences, Qingdao 266071, China; 5The Innovation of Seed Design, Chinese Academy of Sciences, Wuhan 430072, China

**Keywords:** astaxanthin, color variation, binding and transport, lysosome, differentially expressed genes

## Abstract

A mutant of the ridgetail white prawn, which exhibited rare orange-red body color with a higher level of free astaxanthin (ASTX) concentration than that in the wild-type prawn, was obtained in our lab. In order to understand the underlying mechanism for the existence of a high level of free astaxanthin, transcriptome analysis was performed to identify the differentially expressed genes (DEGs) between the mutant and wild-type prawns. A total of 78,224 unigenes were obtained, and 1863 were identified as DEGs, in which 902 unigenes showed higher expression levels, while 961 unigenes presented lower expression levels in the mutant in comparison with the wild-type prawns. Based on Gene Ontology analysis and Kyoto Encyclopedia of Genes and Genomes analysis, as well as further investigation of annotated DEGs, we found that the biological processes related to astaxanthin binding, transport, and metabolism presented significant differences between the mutant and the wild-type prawns. Some genes related to these processes, including crustacyanin, apolipoprotein D (ApoD), cathepsin, and cuticle proteins, were identified as DEGs between the two types of prawns. These data may provide important information for us to understand the molecular mechanism of the existence of a high level of free astaxanthin in the prawn.

## 1. Introduction

Carotenoids are tetraterpene pigments that can form a series of colors such as red, orange, and yellow throughout organisms from fungus to vertebrates. A total of 850 carotenoids have been found in nature, of which about 50 are carotenes and 800 are xanthophylls [1,2]. Carotenes are hydrocarbons, including α-carotene, β-carotene, and so on, while xanthophylls, such as lutein, zeaxanthin, and astaxanthin, contain oxygen atoms such as hydroxy [3]. Carotenoids are important nutrients for animals, acting as vitamin A providers and antioxidants [4], with essential effects on ovarian maturation and growth [5,6]. In addition, the presence of carotenoids in the feathers, skin, and muscles always endows the animals with carotenoid-based colors [3,7,8]. Carotenoids exhibit diverse forms in animals. For example, β-carotene, β-cryptoxanthin, lutein, and astaxanthin can be accumulated in birds [7,9,10]. β-carotene, zeaxanthin, 3-hydroxyechinenone, and astaxanthin are found in some insects [11,12]. Astaxanthin is enriched in salmonid flesh [13]. Pectenolone is in a rare variant of the shellfish Yesso scallop [14]. In crustaceans, astaxanthin is the main carotenoid and accumulated in the shell, the epidermis of crabs, and prawns [15,16,17].

However, carotenoids cannot be synthesized de novo in animals, which are usually accumulated from diets or modified through metabolic reactions. The accumulation of carotenoids in animals involves a series of processes, such as absorption, transport, metabolism, binding, and storage [18]. Usually, the intestine and stomach are important tissues for carotenoid absorption, and the dissolution of carotenoids is the first phase of the process of absorption [19]. Carotenoids can be emulsified into lipid droplets in the stomach, where gastric lipase hydrolyses play important roles [20]. During the process of digestion, carotenoids are incorporated with other lipids into the mixed micelles, which is necessary for their uptake by the enterocyte. Other steps such as uptake in enterocytes and incorporation for transport into chylomicrons are then followed [19,21,22]. Once carotenoids are taken up, they will be metabolized. For animals, hepatopancreas and β-carotene oxygenases are essential metabolic tissues and enzymes. Finally, these carotenoids are metabolized into other metabolites or deposited directly into the skin, eyes or muscles. So far, studies have found that some factors from the environment or the underlying genetic mechanisms may affect the accumulation of carotenoids in animals. Research has proven that a genetic basis, such as some transport proteins or enzymes, plays essential functions in these processes [18,23,24]. Scavenger receptor class B (SRB) can recognize lipoproteins carrying carotenoids, such as lutein, carotene, and zeaxanthin, and then facilitate the delivery of carotenoids into target tissues [25,26,27]. The relationship between SRB and carotenoids uptake has been demonstrated in birds, salmonids, *Drosophila*, and scallops [13,28,29,30]. Lipoproteins such as the high-density lipoprotein (HDL), vitellogenin, and ATP-binding cassette transporter have been demonstrated as carriers that can control the carotenoid transport [31,32,33]. Additionally, β-carotene oxygenase is involved in the cleavage of carotenoids, and its mutation results in carotenoid accumulation in chickens and scallops [34,35].

In crustaceans, several genes were found related to carotenoid binding and carotenoid-based colors. Crustacyanin, a carotenoid-binding protein that was isolated from the carapace of the lobster *Homarus gammarus*, was also associated with astaxanthin binding [36]. The original color of astaxanthin is orangish red, and combining astaxanthin with crustacyanin can increase the variation in color. When boiling, the denaturation of crustacyanin causes the release of astaxanthin, which turns the shell of the crustacean red [37]. In *Fenneropenaeus merguiensis*, sarcoplasmic calcium-binding protein, arginine kinase, and actin were considered to play a role in astaxanthin-based color formation [38]. In *Halocaridina rubra*, the cytochrome P450 family was thought as the possible candidate gene responsible for the production of astaxanthin [39]. In *E. carinicauda*, knockdown of *EcBCO-like6*, encoding carotenoid oxygenases, led to a variation in carotenoid levels and color changes in the hepatopancreas [40].

The wild-type ridgetail white prawn (*E. carinicauda*) is usually in transparent body color, with blue or red spots. Recently, we isolated a mutant with orange-red body color. The orange-red-variant prawns have a significantly higher concentration of astaxanthin compared with the wild-type prawns. Interestingly, astaxanthin in the orange-red prawns is mainly in a free form in the muscles (94%) and waste (60.4%), while it is mainly in an esterified form in the muscles (75.6%) and waste (78.4%) of the wild-type prawns [41], which exhibits a great difference with the known existing form of astaxanthin in crustaceans. However, the underlying mechanism regulating the free astaxanthin enrichment in *E. carinicauda* is not clear. In the present study, comparative transcriptome analysis based on Illumina high-throughput sequencing was performed on the orange-red-variant and wild-type prawns. A number of differentially expressed genes between them were identified and analyzed. The present data provided useful information to further investigate the molecular mechanisms of carotenoid enrichment and carotenoid-based body color variation in crustaceans.

## 2. Materials and Methods

### 2.1. Sample Collection

The orange-red-variant and wild-type prawns (Appendix A) were both cultured in our laboratory. A total of 60 individuals at the intermolt stage, including 30 orange-red-variant prawns and 30 wild-type prawns, were sampled for transcriptome analysis. The prawns at the intermolt stage were selected for experiments according to Gao et al. [42]. Additionally, another 25 orange-red-variant prawns and 25 wild-type prawns at the intermolt stage were collected for quantitative real-time PCR analysis (qPCR). The average body length and body weight of the prawns were 4.38 ± 0.42 cm and 0.98 ± 0.31 g, respectively. The cephalothoraxes of these prawns were dissected and immediately frozen in liquid nitrogen. Intestine was also dissected individually and mixed with cephalothorax. Tissues were stored at −80 °C for total RNA extraction.

### 2.2. RNA Isolation, Sample Pooling, and cDNA Synthesis

Total RNA was extracted separately from samples by RNAiso (TaKaRa, Kyoto, Japan) according to the manufacturer’s instructions. The extracted RNA was then assessed by electrophoresis on 1% agarose gel and quantified by a NanoDrop 2000 spectrophotometer (Thermo Fisher Scientific Inc., Walthman, MA, USA). The total RNA isolated from 30 orange-red-variant prawns were randomly divided into three subgroups. Equivalent total RNA from each sample of one subgroup (containing 10 individuals) was mixed. The total RNA from the wild-type prawns was treated accordingly. Finally, three RNA sample replicates from the orange-red prawns (EcR-1, EcR-2, and EcR-3) and three replicates from the wild prawns (EcW-1, EcW-2, and EcW-3) were prepared for RNA-seq analysis.

For another 25 orange-red-variant prawns, they were randomly divided into five sample replicates with five individuals in each replicate. The 25 wild-type prawns were grouped in the same way; the total RNA from each sample replicate was extracted and assessed as mentioned above. About 1 μg RNA was then used for cDNA generation with the PrimeScript RT Reagent Kit (TaKaRa, Kyoto, Japan). According to the manufacturer’s instructions, the genomic DNA (gDNA) was firstly removed by using a gDNA Eraser, and then the first-strand cDNA was synthesized by a PrimeScript RT Enzyme with random primers.

### 2.3. Library Construction and Sequencing

The construction and sequencing of RNA-seq libraries were conducted according to Illumina’s RNA sample preparation protocols by Gene Denovo Biotechnology Co. (Guangzhou, China). Briefly, the mRNA from each sample was enriched using Oligo(dT) beads. The enriched mRNA was fragmented into short fragments by using fragmentation buffer. After that, these short fragments were used as templates for first-strand cDNA synthesis with random primers. The second-strand cDNA was then synthesized by DNA polymerase I, RNase H, dNTP, and buffer. The cDNA fragments were purified with the QiaQuick PCR Extraction Kit (Qiagen, Duesseldorf, Germany), end repaired, poly(A) added, and ligated to Illumina-sequencing adapters. The ligation products were size selected by agarose gel electrophoresis, PCR amplified, and sequenced using Illumina HiSeqTM 4000.

### 2.4. Assembly and Annotation

The raw reads were filtered by removing adaptors. Low-quality reads containing more than 10% of unknown nucleotides or more than 50% of low-quality bases (*Q* ≤ 10) were removed. The high-quality reads were mapped to a ribosome RNA (rRNA) sequence to remove residual rRNA reads. The clean reads were then assembled into unigenes using the Trinity program, which combines three components: *Inchworm, Chrysalis*, and *Butterfly* [43]. For annotation analysis, unigenes were aligned by BLASTx (*E* < 10^−5^) to databases, including the National Center for Biotechnology Information non-redundant protein (Nr) database (http://www.ncbi.nlm.nih.gov, accessed on 19 November 2018), the Swiss-Prot protein database (http://www.expasy.ch/sprot, accessed on 19 November 2018), the Kyoto Encyclopedia of Genes and Genomes (KEGG) database (http://www.genome.jp/kegg, accessed on 19 November 2018), and the Cluster of Orthologous Groups (COG/KOG) database (http://www.ncbi.nlm.nih.gov/COG, accessed on 19 November 2018). Protein functional annotations could then be obtained according to the best alignment results.

### 2.5. Analysis of Differentially Expressed Unigenes

Gene abundances were calculated and normalized to reads per kb per million (RPKM) reads [44]. To identify differentially expressed genes (DEGs) across samples or groups, the edgeR package was used. Genes with a fold change ≥2 and a false discovery rate (FDR) <0.05 in the two groups were deemed DEGs. DEGs were then subjected to enrichment analysis of GO functions and KEGG pathways. The calculated *p*-value went through FDR correction, taking an FDR ≤0.05 as a threshold. Only GO terms and KEGG pathways meeting this condition could be defined as significantly enriched terms and pathways in DEGs.

### 2.6. Multiple Sequence Alignment and Phylogenetic Analysis

Differentially expressed lipoproteins encoding genes were used for further phylogenetic analysis. Complete open reading frame (ORF) regions and amino acid sequences were deduced using ORFfinder (https://www.ncbi.nlm.nih.gov/orffinder/, accessed on 8 April 2021). Conserved protein domains were predicted with SMART databases (http://smart.embl-heidelberg.de/, accessed on 8 April 2021) [45]. Amino acid sequences of lipoproteins from different species for phylogenetic analysis were obtained from the GenBank™/EBI or UniProt databases [46]. Multiple sequence alignment was performed using ClustalW (https://myhits.sib.swiss/cgi-bin/clustalw, accessed on 8 April 2021), and phylogenetic analysis was performed using MEGA7.0 software [47]. The phylogenetic tree was constructed by the neighbor-joining (NJ) distance algorithm [48]. A bootstrapping test was adopted with 1000 replications.

Mutation analysis was performed for the differentially expressed crustacyanin genes. Clean reads from the wild-type prawns and the orange-red-variant prawns were used to assemble their transcriptome data separately. The sequences of crustacyanin from these two transcriptomes were then extracted. Deduced amino acid sequences were obtained using ORFfinder, and multiple sequence alignment was performed using ClustalW.

### 2.7. Validation by Quantitative Real-Time PCR

In order to verify the accuracy of RNA-seq data, nine unigenes were selected to detect their expression levels in orange-red prawns and wild prawns by quantitative real-time PCR (qPCR) analysis using the Eppendorf Mastercycler ep realplex (Eppendorf, Hamburg, Germany). The 18S rRNA gene was used as an internal reference. Information of the primers is listed in Table 1. The qPCR program was set as follows: one cycle of 94 °C for 2 min, followed by 40 cycles of: 15 s at 95 °C, 15 s at annealing temperature for each pair of primers, and 20 s at 72 °C, and a final 20 min melting curve step was added to test the specificity of these products. All samples were performed in quadruplicates. Relative gene expression levels were calculated following the comparative Ct method with the analysis formula 2-ΔΔCt [49,50,51]. The qPCR results were compared with the transcriptome data to detect the expression correlation of the selected genes.

### 2.8. Statistical Analyses

Statistical significance between orange-red-variant prawns and wild-type prawns was analyzed by using SPSS (version 20). All data obtained from the studies were presented as means ± SE, and the results were analyzed by one-way analysis of variance (ANOVA). Significant differences at *p* < 0.05 were labeled with asterisks.

## 3. Results

### 3.1. Basic Information of the Transcriptome

In total, 76558224, 75407822, 78269646, 84068372, 72076762, and 70924248 raw reads were obtained for each sample by using Illumina HiSeqTM 4000. After filtration, 73335584 (95.79%), 72788448 (96.53%), 75141044 (96.00%), 80832778 (96.15%), 69700338 (96.7%), and 68597552 (96.72%) high-quality clean reads were obtained, respectively (Table 2). The filtered high-quality clean reads were then mapped to the reference transcriptome, and 87.67–88.34% filtered reads were mapped. Finally, these clean reads were assembled and then clustered into 78,224 unigenes with an average length of 824 bp and half of the total assembly (N50) length of 1565 bp. A total of 75,947 existed in the EcR group, and 76,247 existed in the EcW group. In total, 19,461 unigenes were annotated in the database, including 19,285 (24.7%) unigenes with an annotation in the Nr database, 13,808 (17.7%) unigenes in the Swiss-Prot database, 5934 (7.59%) unigenes in the GO database, 12,732 (16.3%) unigenes in the KOG database, and 9150 (11.7%) unigenes in the KEGG database.

### 3.2. Identification and Verification of Differentially Expressed Genes (DEGs)

A total of 1863 unigenes were identified as DEGs between the orange-red prawns (EcR) and the wild-type prawns (EcW), including 902 highly expressed DEGs and 961 low-expressed DEGs in EcR compared with the wild-type prawns. Nine unigenes (Table 1) were selected for quantitative real-time PCR analysis. Among them, Unigene0059050 encoded an esterase, which could hydrolyze esterified compounds. Unigene0036561 was annotated as an apolipoprotein, which might be related to the process of transport. Unigene0054180 was annotated as crustacyanin. Unigene0037069, Unigene0038920, and Unigene0043447 were the genes coding cuticle proteins. These genes might be related to transport and binding functions. The transcriptome data showed that most selected genes showed significantly lower expression levels in the orange-red prawns than in the wild-type prawns (Figure 1A). The qPCR results showed highly consistent expression profiles of these genes compared to the transcriptome data (Figure 1B), which suggested that the transcriptome data were reliable.

### 3.3. GO Term and KEGG Pathway Enrichment Analysis of DEGs

In order to understand which biological processes and signaling pathways might participate in the formation of orange-red color in *E. carinicauda*, GO and KEGG analyses were performed on the DEGs. All DEGs were sorted into 29 functional groups (Figure 2). For the group EcW and EcR, the significant GO molecular function terms were mainly concentrated into categories including “catalytic activity”, “binding”, “transporter activity”, and “structural molecule activity”. Of the biological-process-related genes, most were involved in a “cellular process”, a “single-organism process”, and a “metabolic process”. “Membrane” is the most significantly enriched GO term of cellular components. Results also revealed that the top five terms where DEGs had a much higher expression level in the group EcR included “catalytic activity”, “cellular process”, “binding”, “single-organism process”, and “metabolic process”. Of the low-expressed DEGs, they were mainly enriched in three terms, such as “catalytic activity”, “metabolic process”, and “binding”. In general, “catalytic activity”, “metabolic process”, and “binding” were the terms with the most differentially expressed unigenes.

Only 29 identified DEGs were enriched in pathways by KEGG analysis. The top 20 pathways were shown in Figure 3A. However, most of the enriched pathways only contained one unigene with a *Q*-value greater than 0.1. The “Phagosome” and “Lysosome” were the most significantly enriched pathways (*Q* < 0.05).

### 3.4. Functional Classification of DEGs

In order to further identify genes related to astaxanthin enrichment in orange-red prawns, DEGs with an annotation were categorized according to their reported functions. Among the 1863 DEGs, 279 unigenes were annotated in the Nr database, including 139 highly expressed DEGs (Appendix A) and 140 low-expressed DEGs (Appendix A) in the EcR prawns compared with the wild-type prawns.

Notably, according to the transcriptome analysis, a total of 31 unigenes encoding cuticular proteins or related proteins, which were with similar domains or functional annotations, exhibited higher expression levels in EcW than in EcR (Table 3). These genes were classified into three groups based on their domains. Among them, the deduced amino acid sequences of seven unigenes, including Unigene0017103, Unigene0032773, Unigene0035142, Unigene0037069, Unigene0038920, Unigene0043447 and Unigene0073146 contained a cuticle_1 domain. Unigene0003529, Unigene0019577, Unigene0019752, Unigene0019787, Unigene0022558, Unigene0029471, Unigene0039421, and Unigene00056168 encoded proteins containing a chitin_bind_4 domain, while Unigene0061823 and Unigene0001656 encoded proteins containing a CBM 14 domain.

Binding is an important process for carotenoid metabolism. GO analysis also revealed “binding” was the main term with much more differentially expressed unigenes. Except for cuticular proteins, many other structural genes in the “binding” term, such as tubulin, fibrillin, and fibronectin, were significantly higher in the orange-red prawns compared with those in the wild-type prawns. Additionally, as shown in Table 4, carotenoid-binding protein crustacyanin and hemocyanin genes also have different expression levels between the group EcR and EcW. The result of the sequence alignment revealed that crustacyanin in the wild-type prawns and the orange-red-variant prawns shared 94.85% sequence identity, and both shared a relative similarity with crustacyanin subunit A3 protein in the lobster (Figure 4). Several amino acid residues (residues 6, 87, 92, 104–106, 108, 118–119, 124–125, 129, 131, 134, 144, 191, and 193) showed differences between the EcR and EcW groups. Interestingly, significant changes in the region (PYEVIETDYETYSC) were found, which was the lipocalin consensus region (Figure 4, green colored).

Genes related to transport and metabolism were also analyzed. DEGs including vitellogenin, ApoD, and StAR-related lipid transfer protein were related to the transport process (Table 4). Genes annotated as vitellogenin were all with higher expression levels in the orange-red-variant prawns. Phylogenetic analysis showed that they belonged to apolipocrustacein (apoCr) rather than vitellogenin in crustaceans (Figure 5). By contrast, the three apolipoprotein genes showed different patterns, some of which were with higher expression levels, while the others were with lower expressions in the variant. The result of sequence alignment revealed that Unigene0036561 and Unigene0044620 shared a 65.63% sequence similarity, while they shared 40.61% and 37.31% sequence similarities with Unigene0028810, respectively (Appendix A). Phylogenetic analysis also demonstrated that Unigene0036561 and Unigene0044620 were clustered together, which were all with lower expression levels in the orange-red-variant prawns. In addition, most pathways in metabolism were downregulated in orange-red prawns (Figure 3B). Among them, “Lysosome” was the most significantly enriched pathway, and DEGs such as cathepsin B were included in this pathway (Table 5).

## 4. Discussion

Carotenoid absorption, transport, and metabolism are several crucial steps of carotenoid-dependent coloration. In the present study, the expression profiles of many genes related to carotenoid absorption, transport, metabolism, binding, and deposition were compared between orange-red prawns and wild-type prawns. Lipid digestion and micelle formation are important processes of carotenoid absorption before its uptake in enterocytes [21,52]. After the dissolution of carotenoids, fat-soluble carotenoids are incorporated with other lipids into mixed micelles, which are composed of phospholipids, cholesterol, lipid digestion products, and bile salts [53]. It seems that only free forms of carotenoids are absorbed by the intestinal mucosa, suggesting that esterified forms are first hydrolyzed, and esterification may affect the absorption efficiency [19,54]. The enzymes involved in the hydrolysis of carotenoid esters are probably lipase and esterase, as they have been shown to hydrolyze carotenoid esters [55]. However, genes encoding these kinds of enzymes were hardly found in the DEGs of the present transcriptome, except for Unigene0059050, which encoded a carboxylesterase (esterase FE4) and exhibited higher expression levels in EcW than in EcR. In addition to carotenoid absorption, previous studies have suggested that the incorporation of carotenoids into the carotenoid-binding protein complex and subsequent deposition require a step for the hydrolysis of carotenoid esters [16]. Therefore, the transcriptome data showed that there was no obvious difference in the carotenoid absorption process between EcW and EcR, and the high expression of one carboxylesterase encoding gene in EcW might be related to other carotenoid metabolic processes.

The lipophilicity of carotenoids determines that their transport to various tissues via circulatory and lymphatic systems is intimately linked to lipoproteins. Once loaded with carotenoids, apolipoproteins, components of lipoproteins, can help absorbed carotenoids target various tissues [56,57,58]. Apolipoprotein D (ApoD), a member of the lipocalin family, was once considered a candidate gene associated with carotenoid transport [59]. In *Macrobrachium rosenbergii*, knockdown of the MLC gene, a member of the lipocalin family gene, resulted in a body color shift from blue to orange-red [60]. Additionally, astaxanthin can be transported by HDL and vitellogenin in the chum salmon [33,61]. StARD3 was also reported to be involved in carotenoid transport [62]. The differential expression of these genes in the orange-red prawns and the wild-type prawns indicated that the transport process might be important for carotenoid enrichment in the orange-red prawns.

Metabolism and utilization in target tissues will greatly contribute to the in vivo accumulation of astaxanthin. Phagocytosis is a process whereby cells engulf large particles, which involves engulfment and internalization [63]. Once a target is bound and the necessary signals are activated, these nascent phagosomes initiate the cellular degradation processes, leading to the eventual destruction of the phagocytosed particle. Usually, there are interactions between phagocytes, lysosomes, and endosomes [63,64]. Lysosome, which contains more than 60 enzymes, is the main organelle for substance degradation and metabolism [65]. In the orange-red prawns, the “lysosome” pathway was mainly enriched with related genes at lower expression levels compared with those in the wild-type prawns. Compared with these two pathways in the wild-type prawns, differences between “phagocytosis” and “lysosome” processes might affect the utilization level and accumulation of astaxanthin in the orange-red prawns, which would also contribute to the orange-red body color of the orange-red prawns.

Another process, the binding of carotenoids to a specific protein, is usually related to body coloration and an essential step for carotenoid deposition. In the present study, plenty of genes related to carotenoid binding were identified as DEGs in the transcriptome data. In invertebrates, crustacyanin is proven to bind and stabilize astaxanthin by forming carotene–protein complexes [66]. Astaxanthin is in the free state within the protein complex [67,68]. Once the connection between them is broken, astaxanthin will dissociate from crustacyanin and reveal its natural color. A lower expression level of crustacyanin might lead to a limitation in the binding and subsequent utilization of free astaxanthin in the orange-red prawns. Furthermore, we found that several amino acid residues in the lipocalin consensus region (PYEVIETDYETYSC) were different in a crustacyanin between the wild-type and orange-red-variant prawns. In the crustacyanin homolog A3 protein, several β strands, which are the components of β sheets, are included in this region, which is essential for carotenoid binding [68]. This indicated that a change of these residues might affect the structure and function of the crustacyanin lipocalin domain, leading to a different astaxanthin-binding ability in the orange-red-variant prawns. However, it still needs further investigation to reveal whether they contribute to the binding function with astaxanthin. Additionally, a group of genes encoding cuticular proteins or related proteins showed lower expression levels in the orange-red prawns compared with those in the wild-type prawns. Cuticle proteins are a kind of structural protein involved in the formation of the crustacean exoskeleton. These proteins bind to chitin and play functions in cuticle calcification through different types of domains, including cuticle_1, chitin_bind_4, and CBM 14 [69]. Cuticle proteins also protect insects from adverse environmental stresses, such as UV, insecticides, and infection by pathogens [70,71]. Additionally, in the silkworm, deficiency of one cuticle protein, *BmorCPH24*, could disrupt Wnt1 function and lead to abnormal body coloration [72]. A lower expression of cuticular proteins in the orange-red prawns might affect their body color by influencing other pigment-related genes, pathways, or the layer of pigment cells in the cuticle.

## 5. Conclusions

In conclusion, the present study identified some genes that differentially expressed between the orange-red prawns and the wild-type prawns through comparative transcriptome analysis. GO term enrichment analysis and KEGG pathway enrichment analysis, as well as further investigation of annotated DEGs, revealed the main differences in the biological processes related to astaxanthin binding and transport and metabolism between two types of prawns. The present data provided insights into understanding the molecular mechanism of the body coloration of the orange-red prawns.

## Figures and Tables

**Figure 1 genes-12-00618-f001:**
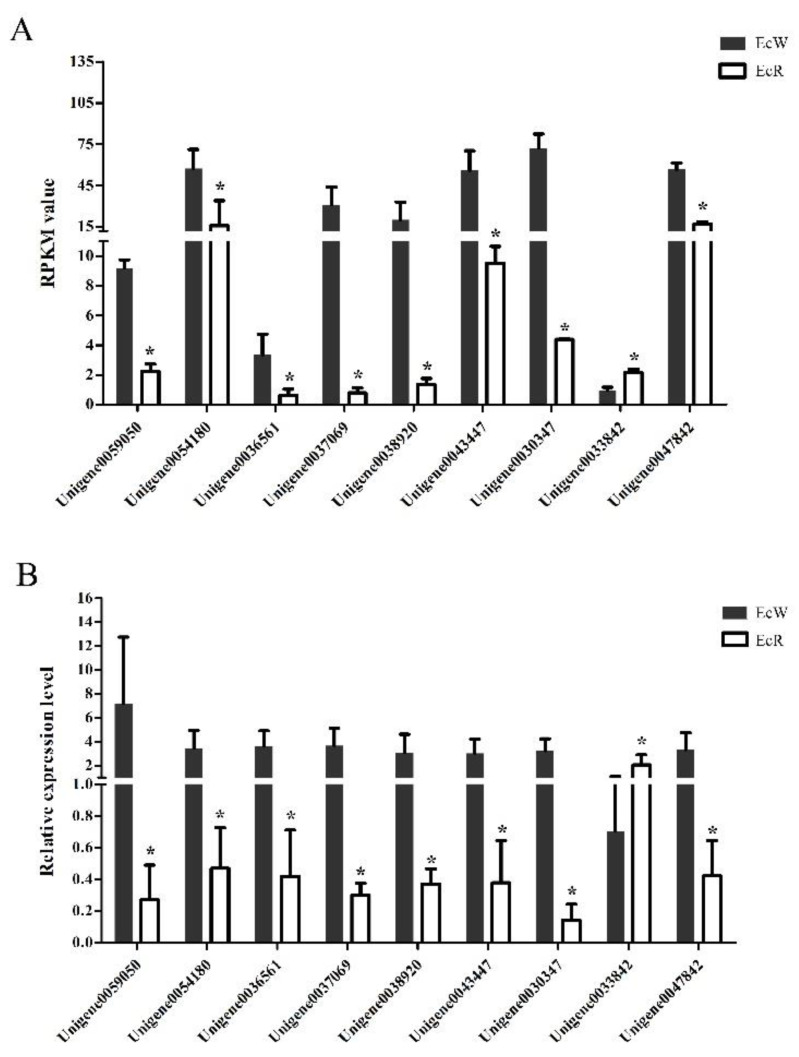
Expression profiles of nine selected unigenes between EcR and EcW prawns. The *x*-axis represents the names of selected unigenes. Columns represent the RPKM value from the transcriptome result (**A**) and the means of relative expression levels from the qPCR result (**B**). The significant difference in the expression levels between the EcR group and the EcW group was labeled with asterisks (*) at *p* < 0.05.

**Figure 2 genes-12-00618-f002:**
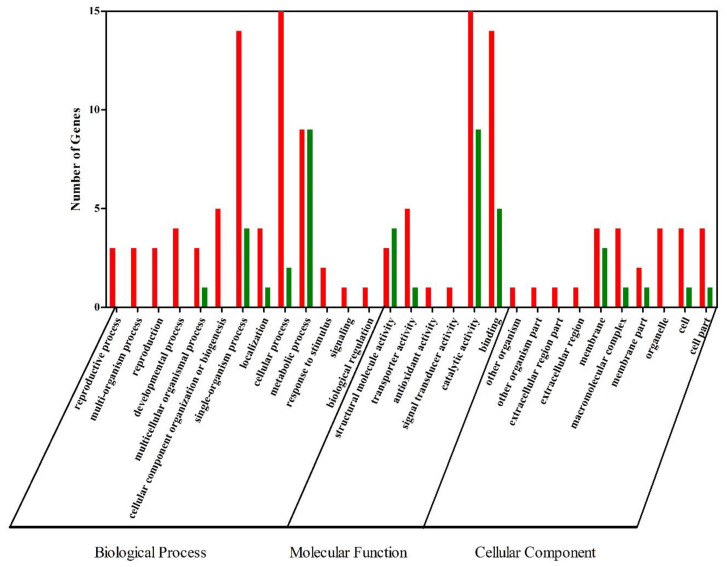
GO annotation of DEGs. DEGs with a GO annotation were divided into three major categories: biological process, cellular component, and molecular function. Red columns represented the DEGs with higher expression levels in the EcR group, while green columns represented the DEGs with lower expression levels in the EcR group.

**Figure 3 genes-12-00618-f003:**
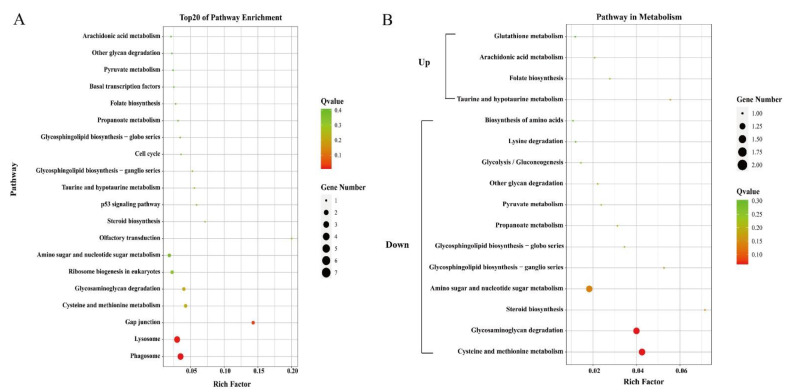
Top 20 enriched pathways (**A**) and pathways related to metabolism (**B**). The circle size and filled portions represented the number of differentially expressed genes in each pathway. The statistical significance was colored according to *Q*-values.

**Figure 4 genes-12-00618-f004:**
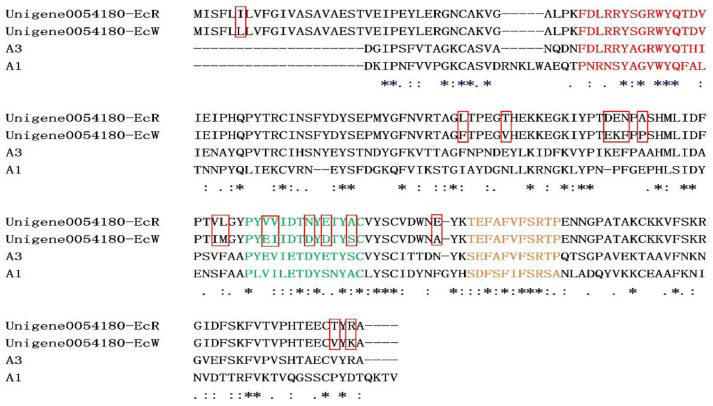
Alignment of crustacyanin amino acid sequences. Unigene0054180-EcR and Unigene0054180-EcW represented the crustacyanin in the orange-red-variant prawns and the wild-type prawns. A1 and A3 represented the crustacyanin subunit proteins in the lobster. Amino acid sequences of lipocalin consensus regions were colored, and different amino acid residues between the orange-red-variant prawns and the wild-type prawns were highlighted in the red box. Asterisks (*) indicated positions which had a single, fully conserved residue, while one black dot (.) and two black dots (:) indicated that the residue sites were less conserved but their physicochemical properties were similar or very similar.

**Figure 5 genes-12-00618-f005:**
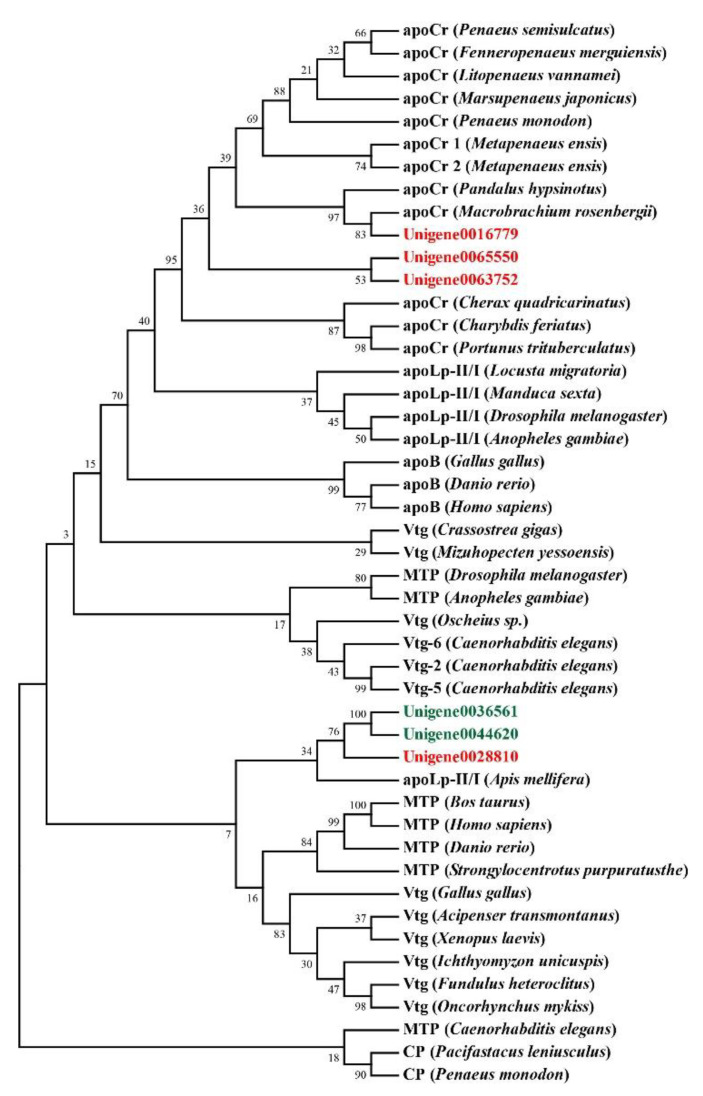
Evolutionary relationships of lipoproteins from *E. carinicauda* and other species. All DEGs with higher expression levels in the orange-red-variant prawns were marked with red font in the tree, while DEGs with lower expression levels were marked with green font. apoCr, apolipocrustacein; MTP, microsomal triglyceride transfer protein; apoB, vertebrate apolipoprotein B; Vtg, vitellogenin; apoLp-II/I, insect apolipophorin II/I precursor. Accession numbers for these sequences were indicated in Appendix A.

**Table 1 genes-12-00618-t001:** Information of primers used for real-time PCR.

Gene ID	Description	Sequence (5′ to 3′)	Tm (°C)
Unigene0059050	Esterase FE4	F: GTTGGACCAAATTTCGGCTCTT R: TGGAGAGAGCGCATGAAAGTTA	59
Unigene0054180	Crustacyanin subunit A	F: TCCTGAGGGAACGCACGAGA R: TGGAGAAGACGAAGGCGAACT	58
Unigene0036561	Apolipoprotein D	F: CTGGGCACAGATTACGAGAACT R: GGTGGGTACGATGGAAGAGG	64
Unigene0037069	Cuticular protein 34	F: CAGAAGCCACGAGCCAGAAG R: GGGTCCTACGAGGATGATGTTAT	57
Unigene0038920	Cuticle proprotein	F: CGTAACTGGCACCTGGATGA R: AGAGGATAGCGTGGGCTGGA	59
Unigene0043447	Cuticular protein 34	F: TCGTCAGAGCCGATGGAAAC R: TGGGAGTGCCATCATCAGTG	58
Unigene0030347	None	F: CTATGCGAATGAATAAGATGAGGAG R: CCAGCGTACCAAGTAATACTGAAA	57
Unigene0033842	Retrovirus-related Pol polyprotein	F: CGATTCACTGTCCCCACTACTC R: GTCTATTTCCTTGATGCTCTTACCA	56
Unigene0047842	Lathosterol oxidase	F: AGTGTAAGAATCCACCAATGCC R: GCCAAGTTTCCAATGAGACAGC	58
18S rRNA	18S rRNA	F: TATACGCTAGTGGAGCTGGAA R: GGGGAGGTAGTGACGAAAAAT	55

Note: F—forward primer; R—reverse primer.

**Table 2 genes-12-00618-t002:** Summary of sequencing and assembly of the transcriptome.

Sample	Raw Reads	Clean Reads	Mapping Ratio	Unigene Number	Ratio
EcW-1	76558224	73335584	64071854 (87.71%)	73,576	94.06%
EcW-2	75407822	72788448	63707819 (87.67%)	74,174	94.82%
EcW-3	78269646	75141044	66104983 (88.14%)	74,006	94.61%
EcR-1	84068372	80832778	71088041 (88.17%)	73,697	94.21%
EcR-2	72076762	69700338	60751470 (87.70%)	73,497	93.96%
EcR-3	70924248	68597552	60474700 (88.34%)	72,320	92.45%
EcW	230235692	221265076	193884656 (87.84%)	76,247	97.47%
EcR	227069382	219130668	192314211 (88.07%)	75,947	97.09%
All	457305074	440395744	387591019 (88.01%)	77,079	98.54%

Note: EcW: the wild prawn group, consisting of three replicates (EcW-1, EcW-2, and EcW-3). EcR: the orange-red prawn group, consisting of three replicates (EcR-1, EcR-2, and EcR-3). Mapping Ratio = Mapped Reads number/the rRNA-removed high-quality clean reads. Ratio = gene number expressed in each sample/all reference gene numbers (78,224).

**Table 3 genes-12-00618-t003:** Information of cuticular proteins in DEGs.

Gene	log2 Ratio (EcR/EcW)	Description	Species
Unigene0039421	−2.98	Calcification-associated peptide−2	*Procambarus clarkii*
Unigene0037661	−2.56	Calcification-associated peptide−2	*P. clarkii*
Unigene0019577	−6.61	Calcified cuticle protein CP14.1	*Callinectes sapidus*
Unigene0010114	−3.86	Calcified cuticle protein CP19.0 isoform A	*C. sapidus*
Unigene0072453	−6.58	Calcified cuticle protein CP19.0 isoform B	*C. sapidus*
Unigene0042544	−5.85	Calcified cuticle protein CP19.0 isoform B	*C. sapidus*
Unigene0038920	−3.89	Cuticle proprotein, partial	*Palaemon varians*
Unigene0032773	−5.13	Cuticle protein 1	*Cherax quadricarinatus*
Unigene0056168	−4.36	Cuticle protein 18.6, isoform B	*Lepeophtheirus salmonis*
Unigene0016948	−4.23	Cuticle protein BD2	*Portunus pelagicus*
Unigene0035595	−2.25	Cuticle protein, partial	*Daphnia magna*
Unigene0003529	−1.74	Cuticular protein	*Tenebrio molitor*
Unigene0073146	−9.77	Cuticular protein 34	*Eriocheir sinensis*
Unigene0035142	−5.96	Cuticular protein 34	*E. sinensis*
Unigene0037069	−5.30	Cuticular protein 34	*E. sinensis*
Unigene0017103	−3.76	Cuticular protein 34	*E. sinensis*
Unigene0043447	−2.56	Cuticular protein 34	*E. sinensis*
Unigene0061823	−1.05	Cuticular protein analogous to peritrophins 1-F, partial	*D. magna*
Unigene0019752	−1.54	Cuticular protein RR-2 motif 78 precursor	*Bombyx mori*
Unigene0029471	−2.05	DD5	*Marsupenaeus japonicus*
Unigene0019787	−1.97	DD5	*M. japonicus*
Unigene0029470	−1.69	DD5	*M. japonicus*
Unigene0044640	−2.05	DD9A, partial	*M. japonicus*
Unigene0001656	−1.55	Obstructor B	*Locusta migratoria*
Unigene0050714	−1.99	Post-molt protein 1	*C. quadricarinatus*
Unigene0022558	−1.80	Strongly chitin-binding protein-1	*P. clarkii*
Unigene0046800	−1.19	Pupal cuticle protein	*Cephus cinctus*
Unigene0008712	−1.12	Pupal cuticle protein	*C. cinctus*
Unigene0023218	−1.42	Cuticle protein 6	*Blaberus craniifer*
Unigene0064526	−1.29	Cuticle protein 6	*B. craniifer*
Unigene0025406	−3.35	Cuticle protein CP1876	*Cancer pagurus*

**Table 4 genes-12-00618-t004:** Transport- and binding-related DEGs.

Functional Categories	GeneID	log2 Ratio (EcR/EcW)	Description	Species
Transport	Unigene0065550	1.57	Vitellogenin 2	*Pandalopsis japonica*
	Unigene0063752	3.65	Vitellogenin	*Pandalus hypsinotus*
	Unigene0016779	1.04	Vitellogenin	*E. carinicauda*
	Unigene0002542	1.52	StAR-related lipid transfer protein 3	*Penaeus monodon*
	Unigene0028810	3.54	Apolipoprotein D	*Zootermopsis nevadensis*
	Unigene0036561	−2.46	Apolipoprotein D	*D. magna*
	Unigene0044620	−1.26	Apolipoprotein D	*D. magna*
Binding	Unigene0029918	2.56	Tubulin	*Tetrahymena thermophila*
	Unigene0073421	2.13	Tubulin/FtsZ family	*T. thermophila*
	Unigene0068154	1.79	Tubulin α-1 chain	*Trichinella murrelli*
	Unigene0013764	1.07	Tubulin α-1 chain-like	*Halyomorpha halys*
	Unigene0026012	1.92	Fibronectin-like	*Ciona intestinalis*
	Unigene0022024	1.53	Fibrillin-1	*Exaiptasia pallida*
	Unigene0036597	1.06	Hemocyanin	*E. carinicauda*
	Unigene0008214	1.15	Hemocyanin	*E. carinicauda*
	Unigene0027904	−1.80	Hemocyanin	*Macrobrachium nipponense*
	Unigene0054180	−1.85	Crustacyanin subunit A	*F. merguiensis*

**Table 5 genes-12-00618-t005:** Differentially expressed genes in Lysosome pathway.

GeneID	log2 Ratio (EcR/EcW)	Description	Species
Unigene0033404	1.02	Cathepsin B	*Pandalus borealis*
Unigene0049424	−1.25	Crustapain	*Pandalus borealis*
Unigene0018100	−2.52	Cathepsin L2	*Litopenaeus vannamei*
Unigene0024515	2.31	Cathepsin L	*D. magna*
Unigene0066106	−3.44	Heparan-α-glucosaminide N-acetyltransferase	*D. magna*
Unigene0022872	−1.11	N-acetyl-β-d-glucosaminidase	*M. nipponense*
Unigene0017001	−2.85	Ecdysteroid regulated-like protein	*Litopenaeus vannamei*

## Data Availability

The data presented in this study are available on request from the corresponding author.

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
