# Peer review of "Transcriptome Analysis Provides Insights into the Mechanism of Astaxanthin Enrichment in a Mutant of the Ridgetail White Prawn Exopalaemon carinicauda"

_genes, 2021, doi:10.3390/genes12050618_

Round 1

Reviewer 1 Report

  1. In line 143-145, the PCA data embed in R package models was mentioned in the section of materials and methods, but the data and analysis does not show in the result section. Explain.
  2. In line 193, to evaluate the quality of transcriptome data, authors are highly subjective to select six DEGs (related to transport and binding function) for qRT-PCR analysis, but it should be randomly selected from NGS data.
  3. The text in all figure should be clearer, and Figure 1 seems unnecessary.
  4. Through NGS data, the authors found the lysosome and phagosome pathway were significantly enriched in Fig. 3. Why the gap junction pathway is ignored ? It would be interesting to see whether the gap junction pathway involved in the process of astaxanthin enrichment.

Author Response

Thanks for the reviewer's constructive suggestions. Please see the point-to-point responses in uploaded word file.

Reviewer 2 Report

I found your study very interesting, with  excellent technical development, 
I sent you some suggestions to make minor corrections before its publication. On the other hand, 
I think it would be interesting to do a taxonomic study to clarify if the variant that you studied is a species that has not been classified.

Lines 21 and 22 in the Abstract; lines 192 and 193 in Results and lines 204,205 and 206 in  Figure 1
In these lines, they refer to the same data, but if we compare the three texts they do not express the same idea. 

Lines 181, 182, 183, and 184:
Check the total of annotated unigenes, 17315 unigenes are missing. 

LINE 203:
Change to EcR Vs EcW to match the order of the columns. 

Line 208:
Change seven to six  

Lines 242 and 243 and lines Lines 345 and 346:
In the text these lines refer to Tables S1 and S2 of the supplementary material, please check if the information corresponds to EcR and EcW prawns or only EcR prawns as expressed in lines 242 and 243.
Table S1. Information of DEGs with higher expression level in EcR prawns
Table S2. Information of DEGs with higher expression level in EcW prawns

Lines 266, 267, and 268:
In my opinion, if these genes are in the table, it is not necessary to repeat them in the text or mention all genes, Cathepsin L2 is missing in the text.

Figure 3. lines 234 and 235
The color of the columns represents DEGs with the higher levels of the two types of prawns studied, so the Up and Down insert in the figure must be eliminated. 

Figure 4:
If possible increase the size of the letters in this figure to make it easier to read. 

In the supplementary material, you could include images of the studied prawns.

Author Response

(The authors gave the same response as above.)

Reviewer 3 Report

This manuscript is about the comparison of transcriptomes between wild type and the oranged-colored ones due to the increased soluble astaxanthin. They compared the transcriptomes in cephalothorax, which was not rather crumsy to explain the physiological phenomenon. I personally, cannot understand the authors intention if they tried to know reason for the phenotypes or its physiological condition. Overall experimental procedures could be changed according to the authors intention. Anyway, they found upregulation in lipoprotein, which may be the key for this manuscript. However, authors should elaborate more about the manuscript as I mentioned below.    

Line 93 describe how to select the shrimp in inter-molt stage

Line 98, metabolites are the products of interplays among various organ. As I explained above, author should explain the interplay among different organs for the astaxanthin metabolism in the introduction section, In this line, authors should justify why they used the cephalothorax for RNAseq study. I guess which includes hepatopancreas but authors should explain about the reason for the choice of the cephalothorax.

Line 204 , title “DiffEXp Gene statistics” in Figure 1 should be properly written. Please do not make any new word for the figure.

Line 273, DEGs in this explain may explain the result of the increased soluble astaxanthin. Therefore all the explanation should focus the physiological responses to the increased soluble astaxanthin. Therefore authors should explain properly about two gene families. First, in table 4 authors found several unigenes, some of which were upregulated while the others were downregulated. Please make multiple alignment and identify which one is which comparing the previous papers below.

Avarre JC, Lubzens E, Babin PJ. Apolipocrustacein, formerly vitellogenin, is the major egg yolk precursor protein in decapod crustaceans and is homologous to insect apolipophorin II/I and vertebrate apolipoprotein B. BMC Evol Biol. 2007;7:3. Published 2007 Jan 22. doi:10.1186/1471-2148-7-3

Line266 where is table 4b I cannot find it.

Line 320 this would be related to table 4B and I could not find the table.  

line 318, Authors have focused only on the transcriptional level, which may not explain properly for the increased soluble astaxanthin in the hemolymph of the shrimp. Author should make the multiple alignments of crustacyanin and its relatives such as other lipocalins to know if mutation at the cartenoid-binding sites. Please read carefully the paper below and explain the comparison of those proteins to understand the cause of the soluble astaxanthin in the shrimp.  

The molecular basis of the coloration mechanism in lobster shell: β-Crustacyanin at 3.2-Å resolution

Michele Cianci, Pierre J. Rizkallah, Andrzej Olczak, James Raftery, Naomi E. Chayen, Peter F. Zagalsky, John R. Helliwell

Author Response

(The authors gave the same response as above.)

Round 2

Reviewer 3 Report

I believe authors  significantly improved  the manuscript according to my suggestions and comments. However, I found a misinterpretation of multiple alignment file.  In Figure S1 I found that critical amino acid replacement in crustacyanin (Unigene0054180). Although authors commented that there is no significant difference in amino acid residues among compared crustacyanins, I found that the a significant change of amino acid in the region (PYEVIETDYETYSC). While the others showed similar amino acid residues in the region, the conserved acidic residue (D) has changed to basic one (N) in the unigene0054180. Author should comment this change and this figure should go to main text not supplementary.    

Author Response

Thanks very much for the reviewer's suggestion. In the Results of the revised manuscript, we have emphasized the difference of specific amino acid residues in the crustacyanin encoded by unigene0054180 of EcR and EcW. This difference was also discussed in the revised manuscript. The related figure has been insert in the main text. Thanks.